# Cytotoxicity of Extracts from *Petiveria alliacea* Leaves on Yeast

**DOI:** 10.3390/plants11233263

**Published:** 2022-11-27

**Authors:** Bruna B. F. Cal, Luana B. N. Araújo, Brenno M. Nunes, Claudia R. da Silva, Marcia B. N. Oliveira, Bianka O. Soares, Alvaro A. C. Leitão, Marcelo de Pádula, Debora Nascimento, Douglas S. A. Chaves, Rachel F. Gagliardi, Flavio J. S. Dantas

**Affiliations:** 1Departamento de Biofísica e Biometria, Universidade do Estado do Rio de Janeiro (UERJ), Rio de Janeiro 20551-030, Brazil; 2Núcleo de Biotecnologia Vegetal, Universidade do Estado do Rio de Janeiro (UERJ), Rio de Janeiro 20550-013, Brazil; 3Instituto de Biofísica Carlos Chagas Filho, Universidade Federal do Rio de Janeiro (UFRJ), Rio de Janeiro 21941-902, Brazil; 4Laboratório de Microbiologia e Avaliação Genotóxica (LAMIAG), Departamento de Análises Clínicas e Toxicológicas, Faculdade de Farmácia, Universidade Federal do Rio de Janeiro (UFRJ), Rio de Janeiro 21941-902, Brazil; 5Laboratório de Química de Bioativos Naturais, Departamento de Ciências Farmacêuticas, Universidade Federal Rural do Rio de Janeiro (UFRRJ), Rio de Janeiro 23897-000, Brazil

**Keywords:** genotoxicity, mutagenicity, oxidative stress, yeast, *Saccharomyces cerevisiae*

## Abstract

*Petiveria alliacea* L. is a plant used in traditional medicine harboring pharmacological properties with anti-inflammatory, antinociceptive, hypoglycemiant and anesthetic activities. This study assessed the potential cytotoxic, genotoxic and mutagenic effects of ethanolic extract of *P. alliacea* on *Saccharomyces cerevisiae* strains. *S. cerevisiae* FF18733 (wild type) and CD138 (*ogg1*) strains were exposed to fractioned ethanolic extracts of *P. alliacea* in different concentrations. Three experimental assays were performed: cellular inactivation, mutagenesis (canavanine resistance system) and loss of mitochondrial function (petites colonies). The chemical analyses revealed a rich extract with phenolic compounds such as protocatechuic acid, cinnamic and catechin epicatechin. A decreased cell viability in wild-type and *ogg1* strains was demonstrated. All fractions of the extract exerted a mutagenic effect on the *ogg1* strain. Only ethyl acetate and n-butanol fractions increased the rate of petites colonies in the *ogg1* strain, but not in the wild-type strain. The results indicate that fractions of mid-polarity of the ethanolic extract, at the studied concentrations, can induce mutagenicity mediated by oxidative lesions in the mitochondrial and genomic genomes of the ogg1-deficient *S. cerevisiae* strain. These findings indicate that the lesions caused by the fractions of *P. alliacea* ethanolic extract can be mediated by reactive oxygen species and can reach multiple molecular targets to exert their toxicity.

## 1. Introduction

Over the centuries, many plants have been used by humans as medicine. The scientific literature has demonstrated the importance of studying plants that have medicinal properties. These species contain a complex mixture of substances that can exert their action alone or synergistically [1,2,3,4,5]. Plant chemical constituents have been a reputable source in the development of drugs for the treatment of various diseases [6]. However, many plants are toxic, causing deleterious effects on users [7].

*Petiveria alliacea* (Phytolaccaceae) is an herb that occurs in tropical regions of South America, Central America, the Caribbean islands and Mexico [8,9]. This species is a medicinal plant with important pharmacological properties with anti-inflammatory, antinociceptive, hypoglycemic and anesthetic activities, in addition to its wide use in the treatment of epilepsy, anxiety and poor memory [8]. In addition, toxic effects have already been described, whose mechanisms of action require further studies [10].

The phytochemical characterization of *P. alliacea* extracts (ethanolic or aqueous) has already been described and the compounds identified in their extracts include flavonoids, triterpenes, steroids, lipids and organosulfur compounds [8,9,11,12].

Considering the reports on the wide use of the ethanolic extract of *P. alliacea* in the treatment of numerous diseases, and studies aimed at identifying biologically active substances, the overall ensemble can help researchers to unravel the molecular mechanisms induced by the extracts [8,13]. Research carried out to determine the toxicity of medicinal plants is based on different experimental designs and on several in vitro [3,12,14,15] and in vivo models [1,16] which have already been used to assess the toxicity of *P. alliacea*; however, this is the first report using yeasts.

*Saccharomyces cerevisiae* are simple eukaryotes whose genome is significantly conserved and shared with humans [17]; therefore, they have been used to evaluate the ability of different chemical and physical agents to induce genotoxic, cytotoxic or mutagenic lesions [18]. Furthermore, yeast is a useful tool to study the DNA repair mechanism [19].

DNA lesions can be generated by reactive oxygen species (ROS) [20]. ROS-induced DNA damage is implicated in several biological processes, such as mutagenesis and carcinogenesis, as well as other diseases [21]. Base excision repair (BER) is the main repair pathway for nuclear and mitochondrial DNA, being responsible for repairing damaged DNA bases, while its absence will result in cell cycle blockage, increase in DNA double breaks and mitochondrial dysfunction, leading to cell death or fixation of mutations that may constitute an early step in the process of carcinogenesis or neurodegenerative diseases [21,22]. Therefore, in this study, the BER-deficient CD138 (ogg1) strain was used, which shows a deficiency in catalytic activity affecting the ability of cells to repair 8-oxoG oxidative DNA damage [23].

Herein, we used different strategies to assess the cytotoxic and mutagenic mechanisms induced by the ethyl acetate and n-butanol fractions from *P. alliacea* ethanolic extract, which are able to generate oxidative lesions in both mitochondrial and genomic DNA of *S. cerevisiae* strains.

Thus, the objective of this work was to evaluate the toxicity and/or mutagenic activity of *P. alliacea* leaf extracts, using *S. cerevisiae* strains as a model, through three experimental assays: (i) inactivation of cultures, (ii) mutagenesis (canavanine resistance system) and (iii) loss of mitochondrial function (petite colonies).

## 2. Results

### 2.1. HPLC-DAD Analysis

HPLC-DAD analysis led to the identification of eight well-defined peaks, which can be related to phenolic compounds. Compounds with R*_T_* = 1.03 (1, λ*_max_* 259 and 290 nm) and 1.32 min (2, λ*_max_* 259 and 290 nm) should correspond to protocatechuic acid and their derivatives. The compounds R*_T_* = 11.93 (5) and 11.95 min (6) should be related to other phenolics compounds derived from cinnamic or benzoic acids, and the compounds with R*_T_* = 10.50 (3–4), 13.09 (7) and 14.19 min (8) are suggested compounds derived from catechin and/or epicatechin (Figure 1).

### 2.2. Survival Effect of the Petiveria alliacea Ethanolic Extract Fracrtions on S. cerevisiae Strains

The results shown a significant (*p* < 0.001) decrease in the survival fraction of the strains FF18733 (wild) and CD138 (*ogg1*), treated with different concentrations of the extract fractions. Also, differences were observed when compared to the controls, 1% DMSO solution, PBS solution, 10% ethanol solution and 4NQO. The LC50 was determined for the FF18733 strain [hexane (25 μg/mL), dichloromethane (75 μg/mL), ethyl acetate (75 μg/mL) and n-butanol (50 μg/mL)] (Figure 2a), and for the CD 138 strain [hexane (15 μg/mL), dichloromethane (50 μg/mL), ethyl acetate (25 μg/mL) and n-butanol (25 μg/mL)] (Figure 2b).

### 2.3. Mutagenesis Evaluation

*S. cerevisiae* strains were treated with LC_50_ fractions and analyzed by the quantification of CAN1-gene-resistant mutant frequency. Figure 3 shows data from spontaneous and induced mutagenesis by the fractions of the ethanolic extract of *P. alliacea* L. (LC50). There was a significant increase in the rate of mutation induced by the fractions only in CD138 strains when compared with the spontaneous mutation rate (1% DMSO solution, 10% ethanol solution or PBS solution) and 4NQO (positive control), confirming an effective mutagenic activity.

### 2.4. Mitochondrial Function

In this study, TTC color assay was used to investigate the frequency of mitochondrial mutants (petites). After treatment of *S. cerevisiae* with LC_50_ concentrations of *P. alliacea* ethanolic fractions, alteration was observed in respiratory metabolism when CD138 yeast strains were treated with fractions of ethyl acetate and n-butanol (Figure 4). Each experiment was performed using a positive control (4NQO) and negative control (PBS solution, 10% ethanol solution or 1% DMSO solution).

## 3. Discussion

Alkaloids, phenolics, flavonoids and other compounds such as terpenes were identified in *P. alliacea* [24]. In our analysis, we suggest the presence of compounds that can be the same pathway derived from shikimic acid such as catechin, epicatechin and derivatives. Among them, we highlight protocatechuic acid, which in turn is associated with antitumor activity, due to its chemotherapeutic potential; depending on the concentration used, it can act selectively against human tumor cells, due to the reduction of mitochondrial membrane potential, decreased activity of Na^+^-K^+^ ATPase, increase in activated caspases-3 and 8 and DNA fragmentation in lung, liver, cervix, breast and prostate cancer cells, promoting non-cancerous cell survival in lung, breast and prostate without causing membrane damage [25,26]. Subsequent mass spectrometry studies with the standards will confirm our hypothesis.

Yeast cell studies can represent a valuable screening tool for detection of genotoxic and mutagenic compounds due to the cost and simplicity of manipulating these cells [27]. Of the ~6000 genes in *S. cerevisiae*, it is estimated that ~60% have a homologue in humans [28]. Comparison of the yeast and human genomes revealed that 30% of known genes involved in human disease have yeast orthologs (i.e., functional homologs) [29]. For our study, two strains of *S. cerevisiae* were chosen: the wild-type FF18733 strain, which has characteristics such as having the wild-type Ogg1 protein and having mitochondrial DNA (mtDNA) more vulnerable to oxidative damage than nuclear DNA, mainly 8-oxoG [30]; and the mutant strain CD138 (Δogg1). The *OGG1* gene encodes the DNA glycosylase protein *ogg1*, which removes the 8-oxoG (7,8-dihydro-8oxoguanine) and Fapy (2,6-diamino-4-hydroxy-5-(methyl) formamididopyrimidine) lesions from the DNA. These lesions lead to mutagenesis and cell death [4].

In our study with the FF18733 strain, by using four fractions of the ethanolic extract of *P. alliacea* leaves, it was possible to observe a reduction in cell viability in all fractions tested. However, it is possible to observe that the mutant strain was more sensitive to the treatment with the ethanolic extract fractions of *P. alliacea* when compared to the wild strain. *S. cerevisiae* strains have been used in studies to define the toxicity and mutagenicity from plant extracts [30], as the dichloromethane extract of *P. alliacea* roots [31], aqueous extract of *Cassia angustifolia* [32] and essential oil of sage (*Salvia officinalis* L.) [33].

Although it has been previously demonstrated that *P. alliacea* induces genotoxic risks in low concentrations [8], there is a lack of information about the effects of ethanolic preparations in in vitro and in vivo models. In this work, cell inactivation was observed in both strains treated with the ethanolic extract fractions, and the cytotoxic activity of these fractions was shown independent of the concentration. These findings corroborate other investigations in which this plant extract also exhibited cytotoxic activity [8,34,35]. In addition, the CD138 strain was more sensitive to treatment with ethanolic extract fractions when compared to the wild-type strain, suggesting that *P. alliacea* extract displays pro-oxidant action and can lead to cell death (cytotoxicity). It has been shown that *P. alliacea* extract increases the concentration of oxidant radicals, such as superoxide (O_2_^−^) and hydrogen peroxide (H_2_O_2_) [1].

Our results point to oxidative stress as the primary cause for the observed cell survival decrease. Although it is not clear which molecules are responsible for its toxic activity, it has been suggested that flavonoids and derivatives, compounds present in the leaves and stems of this plant, could be implied in this activity. Previous studies demonstrated the pro-oxidant properties of flavonoids, which are generally considered to be antioxidants and anticarcinogens, and suggest a dual role for flavonoids in mutagenesis and carcinogenesis [8,36]. Furthermore, it has been reported that the toxicology of this plant is strongly dependent on extraction procedure [35].

Our results suggest that *P. alliacea* extract fractions are capable of generating oxidative lesions, since a greater loss of cell viability was observed in the *ogg1* strain when compared to the wild type. Such lesions, in turn, are generated by an increase in the amount of reactive oxygen species (ROS). Due to their mutagenic character, the accumulation of oxidative lesions in DNA plays an important role in the process of malignant transformation [36]. It was, therefore, necessary to determine whether these fractions would have the ability to induce mutations. For that, mutagenesis assays were performed.

In the present study, it was observed that the *P. alliacea* extract fractions were not able to induce mutations in the genomic DNA (compared to untreated controls and 1% DMSO) for the *S. cerevisiae* strain FF18733. In the assay with the mutant strain CD138, after treatment with the IC50 of each fraction, an increase in the mutation rate was observed when compared to the control. The highest mutagenesis rate was observed for the butanolic fraction.

The oxidative stress generated by the increase in the rate of reactive oxygen species results in the production of lesions in organelles and biomolecules that are important for cellular metabolism. The loss of biological activity or cell death are the main consequences generated by these lesions [37]. Damage to the mitochondrial genome by genotoxic agents and/or oxidants can be assessed by the percentage of induction of petite colonies. Considering the mutagenesis evaluation, an increase was observed in the frequency of mutations in the ogg1 strain when compared to controls (PBS solution or 1% DMSO solution). These results also suggest the induction of oxidative DNA lesions (8-oxoG type lesions) on yeast nuclear DNA (nDNA) by the components of the fractions. The main consequences of these mutations are C-G: A-T-type transversion [34], which results in the loss of cell viability that could explain the inactivation observed in the concentrations of *P. alliacea* ethanolic extract fractions.

In order to confirm the results from the nDNA mutagenesis assay, a petites colonies assay was performed. We observed an increase in the frequency of petite colonies in only CD138 strains, upon treatment with ethyl acetate and n-butanol fractions. Ogg1 was also implicated in the prevention of petite mutants of *S. cerevisiae* [3], indicating a role of Ogg1 in the avoidance of 8-oxoG in mtDNA. It is probable that components from the ethyl acetate and n-butanol fractions of ethanolic extract may produce lesions in the Rho gene of the mtDNA and, consequently, alter mitochondria activity. These results corroborate research suggesting that the ethanolic extract of *P. alliacea* L. cause target deregulation of mitochondrial activity [2,38,39]. Changes in these organelles lead to cell cycle blockage; when these alterations are not repaired, the process of cell death programmed by apoptosis is triggered [37].

Cell death by apoptosis can be induced by both the intrinsic and the extrinsic pathways. In the first (intrinsic) pathway, the phenomenon of disruption of the cytoskeleton and changes in mitochondria induce apoptosis through the release of cytochrome c from this organelle. Cell death by apoptosis is the ideal type of death for tumor destruction as it does not generate an inflammatory process [37]. Thus, components that are capable of inducing apoptosis are of interest to be used as antitumor agents.

In conclusion, the results of this work indicated that fractions of *P. alliacea* ethanolic extract are toxic and/or mutagenic agents; additionally, their consumption in large quantities may represent a risk of developing health problems in their users. Semi-purified extracts contain a mixture of compounds which might generate toxic effects. The ethyl acetate and n-butanol fractions from *P. alliacea* ethanolic extract are cytotoxic and mutagenic, inducing oxidative lesions in the mitochondrial and genomic DNA of *S. cerevisiae* strains.

Ethyl acetate and butanol are the most promising fractions for substances with antitumor activity to be isolated.

Further studies with fractions of the ethanolic extract, such as phytochemical screening biomonitoring, may provide subsidies for better understanding of these phenomena. This data set and the understanding around the *P. alliacea* extract can ensure greater reliability concerning the use of this plant, along with providing a reference for new scientific studies.

## 4. Materials and Methods

*P. alliacea* leaves out of flowering were collected in Niteroi, Rio de Janeiro, Brazil (22°53′55.95″ S and 43°05′09.37″ W). A voucher specimen (HRJ 10.371) has been deposited in the Herbarium of the Rio de Janeiro State University (HRJ 11.710).

### 4.1. Chemical Agents and Reagents

Agar, yeast nitrogen base w/o amino acids (YNBD), bacto-peptone and yeast extract were purchased from DifcoTM, Sweden. Additionally, glucose, ethanol p.a., triphenyl tetrazolium chloride (TTC) and dimethyl sulfoxide (DMSO) were purchased from Merck, Brazil. 4-Nitroquinoline-1-oxide (4-NQO) and canavanine (Can) were purchased from Sigma, New York, NY, USA. Phosphate-buffered saline (PBS) powder was purchased from Laborclin, Brazil. 4-NQO powder was dissolved in 10% ethanol solution and used as experimental positive control. PBS powder was dissolved in distilled H_2_O, sterilized by autoclaving (20 min, 121 °C), stored at room temperature (PBS solution). This was used to wash cells and as experimental negative control. In this study, 1% DMSO solution (1% DMSO) was 1% in distilled H_2_O.

### 4.2. Preparation of Crude Extract and Fractions

The fresh leaves (3720 g) were washed with water and dried in an oven (45 °C) for 24 h. The dried material was macerated and extracted with 1 L of P.A ethanol for 7 days. The extract was filtered and stored at 5 °C. This process was performed five times. Subsequently, the entire filtrate was subjected to evaporation under reduced pressure at 50 °C (1013.25 MBar) to obtain the crude extract (4 g yield) of which an aliquot of 1g was dissolved in distilled water and mixed with n-hexane in ratio of 1:1 (*v*/*v*). The solution was placed in a separatory funnel, and, after vigorous stirring, the hexane fractions were separated, with this procedure being repeated three times. Subsequently, the residual extract was partitioned using organic solvents of increasing polarity [dichloromethane (1:1), ethyl acetate (1:1) and n-butanol (1:1)] in a similar manner to that described for hexane. The fractions were subjected to evaporation under reduced pressure at 50 °C, resulting in a residue of 0.5712g (hexane fraction), 0.0367 g (dichloromethane fraction), 0.0431 g (ethyl acetate fraction), 0.3001 g (n-butanol fraction).

### 4.3. High Performance Liquid Chromatography

HPLC-DAD qualitative analyses were performed on a Shimadzu LC-20AT with a diode-array SPD-M20A detector using a Luna Phenomenex reverse-phase column C-18 (3 μm, 150 mm, 4 mm). The mobile phase consisted of water adjusted to pH 3.0 with formic acid 0.01% (eluent A) and methanol (eluent B). The *P. alliacea* extract was run for 20 min at 1 mL/min and absorbance was monitored between 200–600 nm. The gradient used in chromatography analysis was reported [40]. The flow elution was 1 mL/min. *P. alliacea* extract sample (10 mg) was dissolved in ultrapure water (1 mL), ultrasonicated (30 min) and filtered on Millipore filter (0.45 μm). The volume of injection was 20 μL.

### 4.4. Yeast Strains, Media, Growth Conditions and Reagents

*S. cerevisiae* parental strains FF18733 (Mat a, his7, leu2, lys1, ura3, trp1) (10) and CD138 (ogg1:TRP1). CD138, an isogenic derivative of FF18733, was grown at 28 °C in YPD medium (1% yeast extract, 1% bacto-peptone, 2% glucose, with 2% agar for plates) or YNBD medium (2% glucose, 0. 7% yeast nitrogen base without amino acids with 2% agar for plates) supplemented with appropriate amino acids and bases. Supplemented YNBD (YNBD) medium lacking arginine but containing canavanine (Sigma) at 60 mg/L was used for the selective growth of canavanine-resistant (CanR) mutants. Yeast cultures were grown to a cell density of ~1 × 10^8^ cells/mL (stationary phase) in YPD medium at 28 °C under shaken conditions. Cells were harvested, washed twice with PBS solution and resuspended in the same solution.

### 4.5. Evaluation Ethanolic Extraction S. cerevisiae Strains Survival

Cultures obtained were treated with different concentrations of the ethanolic extract fractions and incubated (28 °C) under agitation for 60 min. For the control, the cultures were traded with 4NQO (1 μg/mL), 1% DMSO solution or PBS solution. Subsequently, appropriate dilutions of treated cells were performed, and plating (YPD-agar plates) and colonies counted after 2 days at 28 °C [40]. The medial lethal concentration (LC50) obtained in the experiments was confirmed with those obtained in Excel software.

### 4.6. Evaluation of the Mutagenic Potential from Ethanolic Extract

Cultures obtained and their aliquots containing approximately 10^3^ cells were used to inoculate 10 new separate cultures in YPD medium (2 mL). These new cultures were treated in the presence of ethanolic extract fractions, 4NQO (1 μg/mL) or 1% DMSO solution for 60 min at 28 °C. Cell density was measured by plating dilutions on YPD-agar plates and counting the colonies after two days. Canavanine-resistant mutants (CanR) were determined after plating appropriate dilutions on YNBD-agar plates supplemented with canavanine, amino acids (uracil 0, 2%, histidine 2%, leucine 1%, lysine 0.4% and tryptophan 0.2%) and mutants were counted after 3 days at 28 °C [32]. Values are the mean of three isolated experiments (six determinations) with standard deviations (SDs) not exceeding 15% (mean ± SD).

### 4.7. Measurement of Mitochondrial Mutants (Petite Colonies)

Aliquots from stationary growing *S. cerevisiae* cultures were treated in the presence of the fractions (LC50), 4-NQO (1 μg/mL), 1% DMSO, at 28 °C. Cultures treated were spread on YPD-agar plates and incubated at 28 °C for 3–4 days. The phosphate buffer with agar (1%) containing TTC poured onto YPD-agar plates and mitochondrial mutants forming a white colony (petite colonies) were scored by TTC color assay [32]. Values are the mean of three isolated experiments. At least 2500 colonies (total 7500) of each strain were scored to determine petite percentage.

### 4.8. Statistical Analysis

Experiments were performed in triplicate. Data obtained in each experiment were submitted to analysis of variance (ANOVA one-way) and Tukey’s post test with multiple comparisons using the program GraphPad InStat 4.0, adopting a confidence level of 95%. LC50 values obtained in the experiments were confirmed with those determined using Excel^®^ software. Data were presented as mean ± standard deviation (±SD).

## Figures and Tables

**Figure 1 plants-11-03263-f001:**
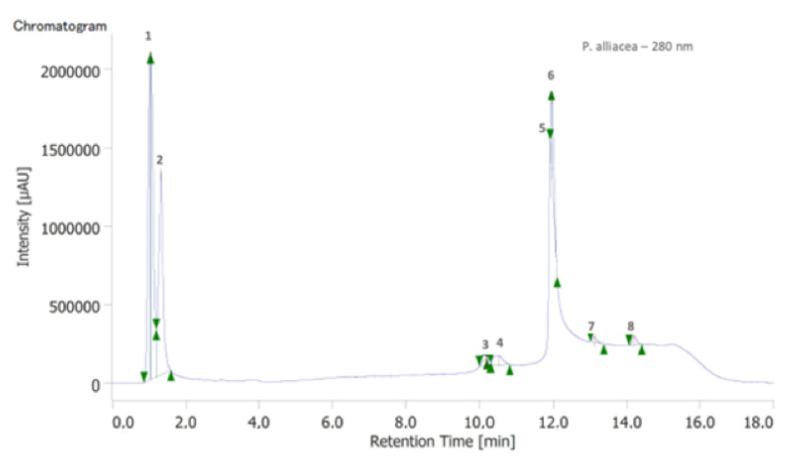
HPLC-DAD—Chromatogram of *Petiveria alliacea* L. (280 nm) extract. 1—protocatechuic acid, 2—protocatechuic acid derivative, 3, 4, 7 and 8 compounds derived from catechin and/or epicatechin. 5 and 6—Derivatives from cinnamic or benzoic acid.

**Figure 2 plants-11-03263-f002:**
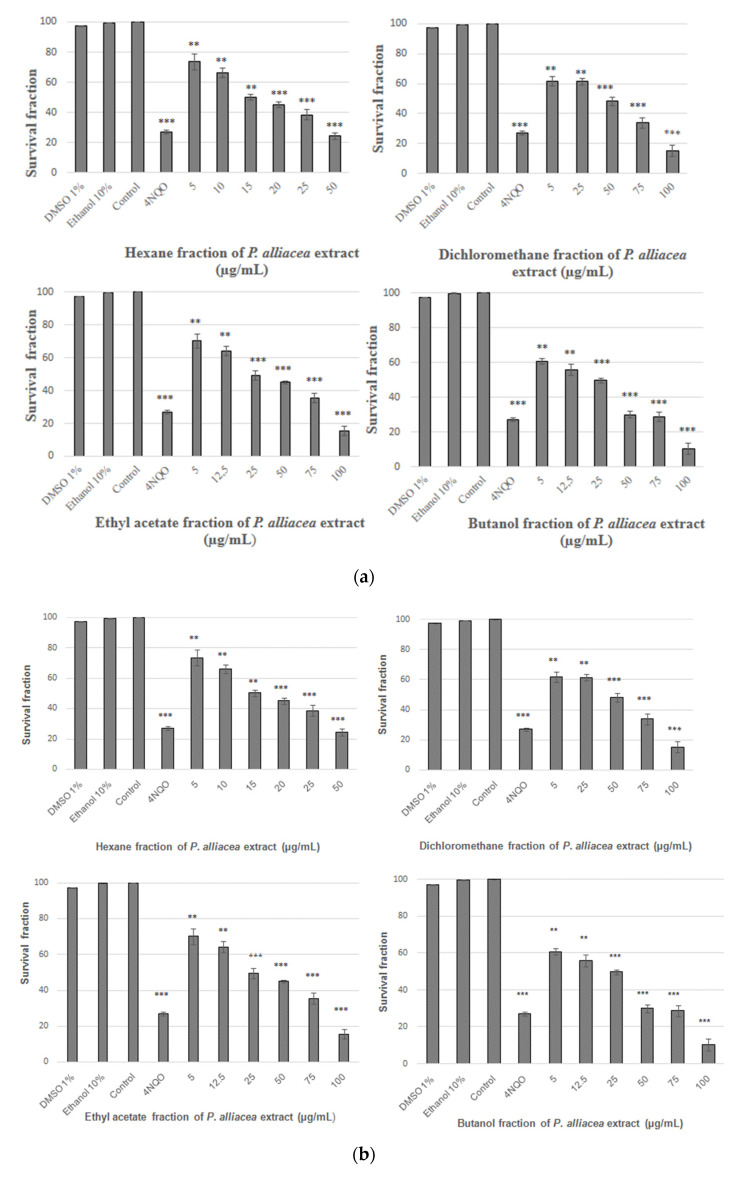
(**a**) Survival fraction of *S. cerevisiae* strain FF18733 submitted to treatment with different concentrations of the *P. alliacea* ethanolic extract of fractions. Evaluation after 60 min of treatment with different concentrations of *P. alliacea* extract fractions. Results express the mean ± standard deviation of three experiments with quintuplicates. ** (*p* < 0.01) *** (*p* < 0.001) vs. control; (**b**) survival fraction of *S. cerevisiae* strain CD138 submitted to treatment with different concentrations of the *P. alliacea* ethanolic extract of fractions. Evaluation after 60 min of treatment with different concentrations of *P. alliacea* extract fractions. Results express the mean ± standard deviation of three experiments with quintuplicates. ** (*p* < 0.01) *** (*p* < 0.001) vs. control.

**Figure 3 plants-11-03263-f003:**
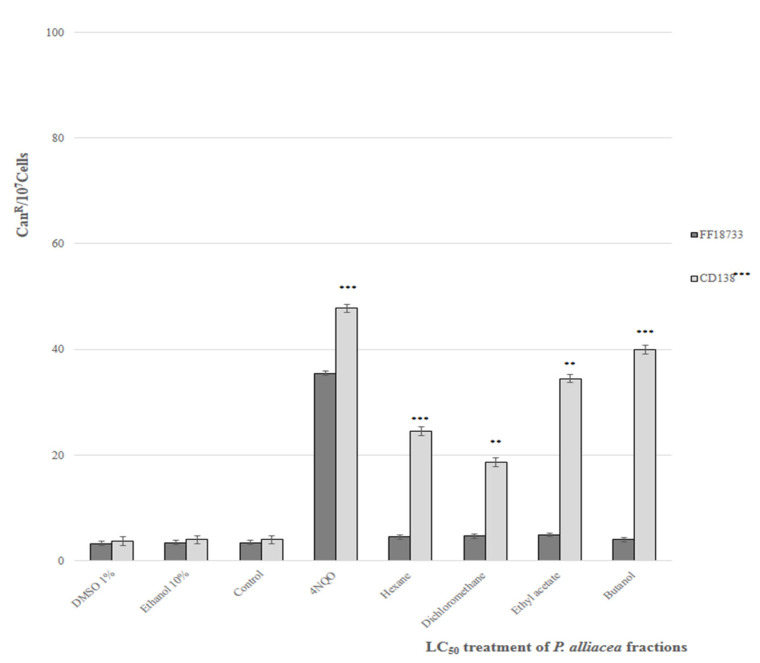
Frequency of spontaneous and exposure-induced mutagenesis in fractions of the *P. alliacea* ethanolic extract of the *S. cerevisiae* strains FF18733 and CD138. Evaluation after 48 h of treatment with LC50 concentrations of the fractions of *P. alliacea* ethanolic extract. Results express the mean ± standard deviation of three experiments with quintuplicates. ** (*p* < 0.01) *** (*p* < 0.001) vs. control.

**Figure 4 plants-11-03263-f004:**
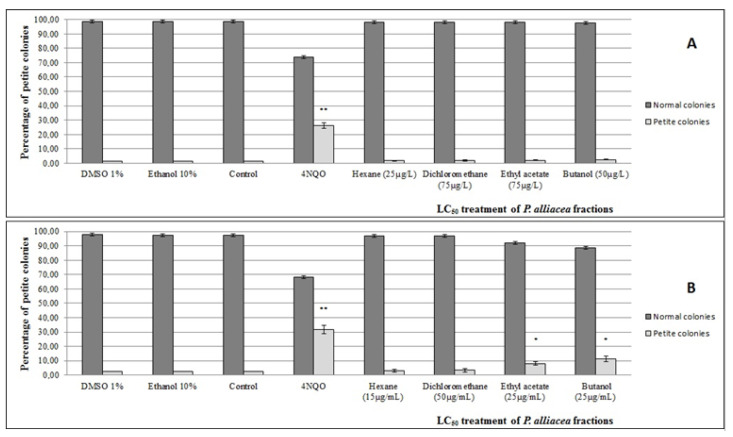
Frequency of petite colonies after cell exposure to fractions of *P. alliacea* ethanolic extract. FF18733 (**A**) and CD138 (**B**) *S. cerevisiae* strains. Evaluation was performed during a 24 h incubation with fractions of *P. alliacea* ethanolic extract and the controls. Results express the mean ± standard deviation of three experiments with quintuplicates. * (*p* < 0.05) ** (*p* < 0.01) vs. control.

## Data Availability

Not applicable.

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
