# Peer review of "Cytotoxicity of Extracts from Petiveria alliacea Leaves on Yeast"

_plants, 2022, doi:10.3390/plants11233263_

Round 1

Reviewer 1 Report

The experimental part of the study was well designed and had proper analytical tools and data analysis. The results obtained are very promising and with the possibility of evaluated the risk of the use Petiveria alliacea L., a plant used in traditional medicine.

I recommend improve text editing (the italics style for scientific name, add spacing before references numbers and several other situations from manuscript).

I suggest improving the representation for LC50 effects, maybe enlarge percentage axis, in Figure 4. 

The Reference paragraph text, in the lines 295-306 is not correct.

Author Response

Manuscript ID: plants-1984451 Submitted to section: Phytochemistry of PLANTS MDPI(Received: 6 October 2022)

Title: Cytotoxicity of extracts from Petiveria alliacea leaves on yeast

Authors: Bruna B.Cal, Luana Araujo, Brenno Nunes, Claudia R. Silva, Marcia Oliveira, Bianka O. Soares, Alvaro Leitão, Marcelo De Padula, Debora Nascimento, Douglas Chaves, Rachel F. Gagliardi, Flavio J. S.Dantas

Rio de Janeiro, November 9, 2022

Dear Editor and  reviewer

Thank you very much for your time and consideration.We are grateful for the decision to suggest resubmission of the manuscript after major revisions.Thank you for the assignments of the reviewers, all corrections and suggestions have been incorporated, as detailed below. The phytochemical evaluation was carried out in the hope of identifying the chemical groups that could be associated with the observed toxic effects. The specific identification, as well as its isolation and purification, remain in perspective for further studies. Statistical analyzes were performed only for the validation of biological tests to assess toxic effects. The manuscript has been revised for according to the referees and revisions to the manuscript were made with the track changes function (attached manuscript).

In response to the general review the English language and style have been revised and corrected. Text intro has been improved and references have been revised, and the list has been updated. The methodology was improved and described in further detail. In the results section, the figures were also improved and the conclusions were based on the results, placed at the end of the discussion section. The conclusion section was suppressed.

We would like to clarify that in this work yeasts were used as the only experimental model because the extracts of this species have already been tested in numerous other models in vivo and in vitro, as has now been pointed out in the introduction. Furthermore, due to the cellular characteristics of yeasts, this model could help to clarify the mechanism of action of the extracts.

Explaining the purpose of the work better, although the literature indicated the cytotoxic and mutagenic effect of extracts of Petiveria alliacea in different experimental models, the aim of this work was to evaluate toxicity and/or mutagenesis activity using S. cerevisiae strains in three experimental assays, selected in order to gather more subsidies to clarify the cytotoxic and mutagenic mechanism of action of the extracts.

And finally considering the content and length of our manuscript, as suggested by Dr.Charles Guo, we changed the manuscript type from article to communication.

Yours sincerely

FlávioJ.S.Dantas

List of corrections indicated by the referee reports:

Review 1 - Responses to reviewers' comments

1.The experimental part of the study was well designed and had proper analytical tools and data analysis. The results obtained are very promising and with the possibility of evaluated the risk of the use Petiveria alliacea L., a plant used in traditional medicine.

Authors' response: We appreciate your positive review.

2.I recommend improve text editing (the italics style for scientific name, add spacing before references numbers and several other situations from manuscript).

Authors' response: The manuscript has been revised and all corrections incorporated.

3.I suggest improving the representation for LC50 effects, maybe enlarge percentage axis, in Figure 4. 

Authors' response: The correction was performed At some points the error bars are not visible due to the low standard deviation.

4.The Reference paragraph text, in the lines 295-306 is not correct.

Authors' response: The mentioned paragraph has been removed

Reviewer 2 Report

Revision of the paper entitled

Cytotoxicity effects induced by fractions of ethanolic extract of 2 Petiveria alliacea L. in Saccharomyces cerevisiae strains

The article describes describes a study on the cytotoxic effect of an interesting biologically active plant. Unfortunately, the results observed on one model do not yield relevant results and the possible cytotoxicity of the extract. Why was only yeast model used for mutagenicity testing?

The text contains a large number of formal errors, they are indicated in the attached text.

The methodologies are vaguely described.

In conclusion, I lack a better explanation of what the point of the study is if the cytotoxicity and mutagenicity results are very clear.

Author Response

Manuscript ID: plants-1984451 Submitted to section: Phytochemistry of PLANTS MDPI(Received: 6 October 2022)

Title: Cytotoxicity of extracts from Petiveria alliacea leaves on yeast

Authors: Bruna B.Cal, Luana Araujo, Brenno Nunes, Claudia R. Silva, Marcia Oliveira, Bianka O. Soares, Alvaro Leitão, Marcelo De Padula, Debora Nascimento, Douglas Chaves, Rachel F. Gagliardi, Flavio J. S.Dantas

Rio de Janeiro, November 9, 2022

Thank you very much for your time and consideration.We are grateful for the decision to suggest resubmission of the manuscript after major revisions.Thank you for the assignments of the reviewers, all corrections and suggestions have been incorporated, as detailed below. The phytochemical evaluation was carried out in the hope of identifying the chemical groups that could be associated with the observed toxic effects. The specific identification, as well as its isolation and purification, remain in perspective for further studies. Statistical analyzes were performed only for the validation of biological tests to assess toxic effects. The manuscript has been revised for according to the referees and revisions to the manuscript were made with the track changes function (attached manuscript).

In response to the general review the English language and style have been revised and corrected. Text intro has been improved and references have been revised, and the list has been updated. The methodology was improved and described in further detail. In the results section, the figures were also improved and the conclusions were based on the results, placed at the end of the discussion section. The conclusion section was suppressed.

We would like to clarify that in this work yeasts were used as the only experimental model because the extracts of this species have already been tested in numerous other models in vivo and in vitro, as has now been pointed out in the introduction. Furthermore, due to the cellular characteristics of yeasts, this model could help to clarify the mechanism of action of the extracts.

Explaining the purpose of the work better, although the literature indicated the cytotoxic and mutagenic effect of extracts of Petiveria alliacea in different experimental models, the aim of this work was to evaluate toxicity and/or mutagenesis activity using S. cerevisiae strains in three experimental assays, selected in order to gather more subsidies to clarify the cytotoxic and mutagenic mechanism of action of the extracts.

And finally considering the content and length of our manuscript, as suggested by Dr.Charles Guo, we changed the manuscript type from article to communication.

Yours sincerely

FlávioJ.S.Dantas

List of corrections indicated by the referee reports:

Review 2

1.The article describes a study on the cytotoxic effect of an interesting biologically active plant. Unfortunately, the results observed in only one model do not bring relevant results and the possible cytotoxicity of the extract. Why was only the yeast model used for mutagenicity testing?

Authors' response: In fact, the effects of extracts of this species have already been evaluated in in vivo and in vitro models, which is the first report using yeasts. As they are eukaryotic cells that share many genes with humans, they are considered a model system for mutagenicity and repair studies, providing more information about the mechanism of action of the substances present in the extracts. The paragraph was rewritten, with the insertion of citations referring to these studies and the references were included in the bibliography. All species names are in italics and grammatical errors have been corrected.

  1. The text contains a large number of formal errors; they are indicated in the attached text.

Authors' response: Errors have all been corrected in the manuscript text.

  1. The methodologies are vaguely described.

Authors' response: The methodologies were described in more detail.

  1. In conclusion, I lack a better explanation of what the purpose of the study is if the cytotoxicity and mutagenicity results are very clear.

Authors' response: Indeed, mutagenic and cytotoxic activity have been previously described in other models. However, in order to fill the gap in relation to the identification of substances responsible for these effects, a phytochemical evaluation of the extracts was associated, using a model still unpublished for these studies.

Reviewer 3 Report

The present manuscript have some interesting data but need extensive revision before its acceptance. Few points are listed below and others are highlighted in the attached text.

1. The whole manuscript needs a serious spell check for typos and other mistakes, like scientific names should be in italics.

2. Only 22 references are cited. To discuss the data properly It requires more recent article citation.

3. Discussion part should be re-frame.

4. In material and methods, authors have described that they performed experiments in triplicate. What does it mean? How many times the whole experiments are repeated with what number of replications?

5. Figure quality is not good enough. In figure 2a, 2b and 3 few bars are without standard deviation bar.

6. On what basis the HPLC peaks are identified?

7. Re-frame conclusion.

Author Response

Manuscript ID: plants-1984451 Submitted to section: Phytochemistry of PLANTS MDPI(Received: 6 October 2022)

Title: Cytotoxicity of extracts from Petiveria alliacea leaves on yeast

Authors: Bruna B.Cal, Luana Araujo, Brenno Nunes, Claudia R. Silva, Marcia Oliveira, Bianka O. Soares, Alvaro Leitão, Marcelo De Padula, Debora Nascimento, Douglas Chaves, Rachel F. Gagliardi, Flavio J. S.Dantas

Rio de Janeiro, November 9, 2022

Dear Editor and reviewer

Thank you very much for your time and consideration.We are grateful for the decision to suggest resubmission of the manuscript after major revisions.Thank you for the assignments of the reviewers, all corrections and suggestions have been incorporated, as detailed below. The phytochemical evaluation was carried out in the hope of identifying the chemical groups that could be associated with the observed toxic effects. The specific identification, as well as its isolation and purification, remain in perspective for further studies. Statistical analyzes were performed only for the validation of biological tests to assess toxic effects. The manuscript has been revised for according to the referees and revisions to the manuscript were made with the track changes function (attached manuscript).

In response to the general review the English language and style have been revised and corrected. Text intro has been improved and references have been revised, and the list has been updated. The methodology was improved and described in further detail. In the results section, the figures were also improved and the conclusions were based on the results, placed at the end of the discussion section. The conclusion section was suppressed.

We would like to clarify that in this work yeasts were used as the only experimental model because the extracts of this species have already been tested in numerous other models in vivo and in vitro, as has now been pointed out in the introduction. Furthermore, due to the cellular characteristics of yeasts, this model could help to clarify the mechanism of action of the extracts.

Explaining the purpose of the work better, although the literature indicated the cytotoxic and mutagenic effect of extracts of Petiveria alliacea in different experimental models, the aim of this work was to evaluate toxicity and/or mutagenesis activity using S. cerevisiae strains in three experimental assays, selected in order to gather more subsidies to clarify the cytotoxic and mutagenic mechanism of action of the extracts.

And finally considering the content and length of our manuscript, as suggested by Dr.Charles Guo, we changed the manuscript type from article to communication.

Yours sincerely

FlávioJ.S.Dantas

List of corrections indicated by the referee reports:

Review 3 - Responses to reviewers' comments

The present manuscript has some interesting data but need extensive revision before its acceptance. Few points are listed below and others are highlighted in the attached text.

1.The whole manuscript needs a serious spell check for typos and other mistakes, like scientific names should be in italics.

Authors' response: The spelling review was performed and other formatting errors were corrected.

  1. Only 22 references are cited. To discuss the data properly It requires more recent article citation.

Authors' response: As the Introduction and Discussion have been reformulated, the number of references has increased to 41.

  1. Discussion part should be re-frame.

Authors' response: The discussion was duly reformulated.

  1. In material and methods, authors have described that they performed experiments in triplicate. What does it mean? How many times the whole experiments are repeated with what number of replications?

Authors' response:  The whole experiments were repeated three times in quintuplicate, resulting in a total of 15 replications.

  1. Figure quality is not good enough. In figure 2a, 2b and 3 few bars are without standard deviation bar.

Authors' response: The figures were edited and their resolution was increased using the PhotoFiltre7 software. At some points the error bars are not visible due to the low standard deviation.

  1. On what basis the HPLC peaks are identified?

Authors' response: The peaks were proposed based on retention time, maximum lambdas identified and shape of the DAD. For the protocatechuic acid we performed the identification by co-injection of the standard;

  1. Re-frame conclusion.

Authors' response: The conclusion has been re-framed and placed at the end of the Discussion

Reviewer 4 Report

This article is interested to citotoxicity of extracts from the leaves of Petiveria alliacea L. Thus the title need to reform. The same for abstract and keywords.

- page 3 (part 2.1 HPLC-DAD analysis): the attribution of peaks is not well justified and remains general. In addition, the extract submitted for analysis is not specified (it is probably the ethanolic extract).

-Why have the authors not tempted themselves to identify and to perform separate tests for each of the main components of Petiveria alliacea L. extrats? The research does not show which component of ethanolic extrat has an effect? Description of analysis of extrats must be given. The article would be vastly improved if the implicated compound itself was identified.  Eg. by extracting and trying the pure compound.

-page 6 (paragraph 4.2): the extraction mode must be clear and well detailed : the use of 1L of ethanol several times is enormous !!. why the authors chose maceration and not another mode of extraction (continuous extraction by soxhlet apparatus, for example): The yield obtained is low.

- No indication on the preparation of the different concentrations tested.

-The choice of the concentration ranges studied is not justified.

-In addition to figures, it is important to present the results in tables.

- Authors have to compare their results with other works and discuss more and more.

- The English language of the manuscript needs to be improved. There are some grammatical errors.

Author Response

Manuscript ID: plants-1984451 Submitted to section: Phytochemistry of PLANTS MDPI(Received: 6 October 2022)

Title: Cytotoxicity of extracts from Petiveria alliacea leaves on yeast

Authors: Bruna B.Cal, Luana Araujo, Brenno Nunes, Claudia R. Silva, Marcia Oliveira, Bianka O. Soares, Alvaro Leitão, Marcelo De Padula, Debora Nascimento, Douglas Chaves, Rachel F. Gagliardi, Flavio J. S.Dantas

Rio de Janeiro, November 9, 2022

Dear Editor and Review,

Thank you very much for your time and consideration.We are grateful for the decision to suggest resubmission of the manuscript after major revisions.Thank you for the assignments of the reviewers, all corrections and suggestions have been incorporated, as detailed below. The phytochemical evaluation was carried out in the hope of identifying the chemical groups that could be associated with the observed toxic effects. The specific identification, as well as its isolation and purification, remain in perspective for further studies. Statistical analyzes were performed only for the validation of biological tests to assess toxic effects. The manuscript has been revised for according to the referees and revisions to the manuscript were made with the track changes function (attached manuscript).

In response to the general review the English language and style have been revised and corrected. Text intro has been improved and references have been revised, and the list has been updated. The methodology was improved and described in further detail. In the results section, the figures were also improved and the conclusions were based on the results, placed at the end of the discussion section. The conclusion section was suppressed.

We would like to clarify that in this work yeasts were used as the only experimental model because the extracts of this species have already been tested in numerous other models in vivo and in vitro, as has now been pointed out in the introduction. Furthermore, due to the cellular characteristics of yeasts, this model could help to clarify the mechanism of action of the extracts.

Explaining the purpose of the work better, although the literature indicated the cytotoxic and mutagenic effect of extracts of Petiveria alliacea in different experimental models, the aim of this work was to evaluate toxicity and/or mutagenesis activity using S. cerevisiae strains in three experimental assays, selected in order to gather more subsidies to clarify the cytotoxic and mutagenic mechanism of action of the extracts.

And finally considering the content and length of our manuscript, as suggested by Section Managing Editor Dr.Charles Guo, we changed the manuscript type from article to communication.

Yours sincerely

FlávioJ.S.Dantas

List of corrections indicated by the referee reports:

Review 4 - Responses to reviewers' comments

1.This article is interested to citotoxicity of extracts from the leaves of Petiveria alliacea L. hus the title need to reform. The same for abstract and keywords.

Authors' response: The title was reformulated, including the suggestion to specify the tissue used to prepare the extracts. The same was done with the abstract and keyword (see attached manuscript).

2.page 3 (part 2.1 HPLC-DAD analysis): the attribution of peaks is not well justified and remains general. In addition, the extract submitted for analysis is not specified (it is probably the ethanolic extract).

Authors' response: The entire study was performed using ethanol extract and its fractions. Regarding peaks, see the answer to question 6 by reviewer 3.

3.Why have the authors not tempted themselves to identify and to perform separate tests for each of the main components of Petiveria alliacea L. extrats? The research does not show which component of ethanolic extrat has an effect? Description of analysis of extrats must be given. The article would be vastly improved if the implicated compound itself was identified.  Eg. by extracting and trying the pure compound.

Authors' response: The work aimed to study the toxic effects using a model in yeast, still unpublished, to confirm these effects and determine the mechanisms of action of the extracts. Phytochemical analysis was performed to try to associate groups of substances present with these toxic effects. Based on the results, protocatechuic acid, cinnamic and catechin epicatechin were the major substances found and could have some implication in these effects, this hypothesis being discussed at the end. The exact identification and isolation of these components constitute an offshoot of this work and will be the object of further studies.

4.page 6 (paragraph 4.2): the extraction mode must be clear and well detailed: the use of 1L of ethanol several times is enormous !!. why the authors chose maceration and not another mode of extraction (continuous extraction by soxhlet apparatus, for example): The yield obtained is low.

Authors' response: The extraction method was rewritten in greater detail in the manuscript. The maceration method was used for its practicality and for being already established in the laboratory. The large amount of alcohol used in the exhaustion method was aimed at extracting the maximum amount of substances present, so far there is no soxlet device available in our laboratory. The yield of the ethanolic extract was 4g for 3,720g of fresh plant material, and this information was included in the manuscript.

5.No indication on the preparation of the different concentrations tested. The choice of the concentration ranges studied is not justified.

Authors' response: The concentrations studied were determined from preliminary tests with crude extract.

6.In addition to figures, it is important to present the results in tables.

Authors' response: Quantitative results are presented in graphs only to avoid repetition.

7.Authors have to compare their results with other works and discuss more and more.

Authors' response: The discussion has been rephrased.

8.The English language of the manuscript needs to be improved. There are some grammatical errors.

Mistakes have been fixed and a language revision has been made.

Round 2

Reviewer 2 Report

No comments.

Reviewer 3 Report

Authors have made sufficient changes in their manuscript. Now it can be accepted for publication.